# GUV long-term measurements of total ozone column and effective cloud transmittance at three Norwegian sites

Tove M. Svendby[1], Bjørn Johnsen[2], Arve Kylling[1], Arne Dahlback[3], Germar H. Bernhard[4], Georg H. Hansen[1], Boyan Petkov[5,6], Vito Vitale[6]

[1]NILU-Norwegian Institute for Air Research, Norway
[2]Norwegian Radiation and Nuclear Safety Authority, Norway
[3]University of Oslo, Norway
[4]Biospherical Instruments, Inc., USA
[5]University G. d'Annunzio, Department of Psychological Sciences, Health and Territory, Italy.
[6]National Research Council, Institute of Polar Sciences (CNR-ISP), Italy

*Correspondence to*: Tove M. Svendby (tms@nilu.no)

## 1. Abstract

Measurements of total ozone column and effective cloud transmittance have been performed since 1995 at the three Norwegian sites Oslo/Kjeller, Andøya/Tromsø and in Ny-Ålesund (Svalbard). These sites are a subset of 9 stations included in the Norwegian UV monitoring network, which uses GUV multi-filter instruments and is operated by DSA and NILU. The network includes unique data sets of high time-resolution measurements that can be used for a broad range of atmospheric and biological exposure studies. Comparison of the 25-year records of GUV (global sky) total ozone measurements with Brewer direct sun (DS) measurements show that the GUVs provide valuable supplements to the more standardized ground-based instruments. The GUVs can fill in missing data and extend the measuring season at sites with reduced staff and/or characterized by harsh environmental conditions, such as Ny-Ålesund. Also, a harmonized GUV can easily be moved to more remote/unmanned locations and provide independent total ozone column datasets. The GUV in Ny-Ålesund captured well the exceptionally large Arctic ozone depletion in March/April 2020, whereas the GUV in Oslo recorded a mini ozone hole in December 2019 with total ozone values below 200 DU. For all the three Norwegian stations there is a slight increase in total ozone from 1995 until today. Measurements of GUV effective cloud transmittance in Ny-Ålesund indicate that there has been a significant change in albedo during the past 25 years, most likely resulting from increased temperatures and Arctic ice melt in the area surrounding Svalbard.

## 2. Introduction

The amount of stratospheric ozone decreased significantly both globally and over Norway during the 1980s and 1990s (WMO 2018; Svendby and Dahlback, 2004). This decrease was mainly caused by the release of ozone depleting substances (ODSs).

In 1987, the Montreal Protocol was signed with the aim of phasing out the production of ODSs. Motivated by this treaty, the
Norwegian Environment Agency established the programme "Monitoring of the atmospheric ozone layer" in 1990. Five years
later, in 1995/1996, the network was expanded and "*The Norwegian UV network*" was established with funding from the
Norwegian Ministry of Health and Care Services and the Norwegian Environment Agency. This network consists of nine
Ground-based UltraViolet (GUV) radiometers located at sites between 58°N and 79°N (Figure 1). The network has been in
operation for 25 years, and the measurements are undertaken by the Norwegian Radiation and Nuclear Safety Authority, DSA
(formerly the Norwegian Radiation Protection Authority, NRPA) and the Norwegian Institute for Air Research (NILU). The
GUV instruments allow the calculation of the UV Index, retrievals of total ozone column, cloud transmittance, and several
other UV-related dose products (Dahlback, 1996; Høiskar et al., 2003; Bernhard et al., 2005). Data and dose products have
been used in several international studies (Bernhard et al, 2013; Bernhard et al, 2015; Schmalwieser et al., 2017; Lakkala et
al., 2020; Bernhard et al, 2020), and the data are available at https://github.com/uvnrpa and Johnsen et al., (2020).

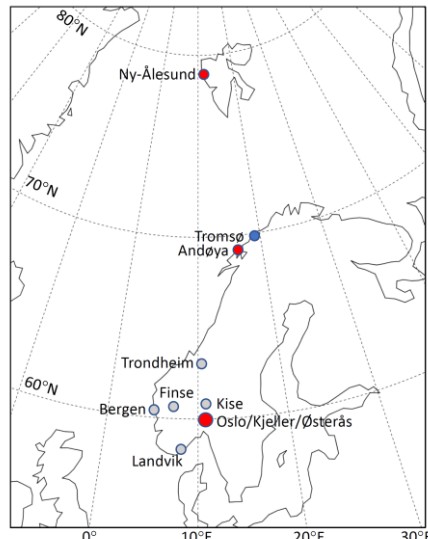

**Figure 1: The Norwegian UV network. Grey circles represent stations operated by DSA, whereas red circles represent sites operated**
**by NILU. The large red circle to the south includes the three stations at Østerås (DSA), Blindern in Oslo, and Kjeller. The instrument**
**in Tromsø (blue circle) was moved to Andøya in 2000.**

The spectral distribution of solar UV radiation reaching the ground depends on the optical properties of the atmosphere, the
solar zenith angle (SZA) and reflection from the Earth's surface. The transmission of solar radiation in the UVB region (280–
315 nm) through the stratosphere is primarily determined by the amount of stratospheric ozone, whereas the attenuation in the
troposphere is mainly due to scattering by air molecules (Rayleigh scattering), aerosols, and clouds. Generally, a decrease in
total ozone column leads to an increase in UVB radiation, assuming no changes in cloudiness or other UV-affecting parameters.

High-wavelength-resolution spectroradiometers can provide detailed information about the spectral distribution of UV radiation. Stamnes et al. (1991) showed that spectra from such instruments can be used to determine total ozone and cloud transmission accurately. However, simpler and cheaper radiometers with channels in both the UVB and the UVA regions, such as the GUVs, have also demonstrated to be a good alternative to expensive spectroradiometers (Dahlback, 1996; Bernhard et al., 2005; Sztipanov et al., 2020).

In this study, we present a 25-year time series of total ozone column (TOC) from the Norwegian UV Network. We have focused on three stations operated by NILU located in Oslo/Kjeller, at Andøya/Tromsø and in Ny-Ålesund as shown by red circles in Figure 1. All stations are equipped with additional total ozone measuring instruments such as Brewer spectrophotometers and a Systeme d'Analyse par Observation Zenithale (SAOZ) instrument. TOCs derived from the GUVs are compared to measurements from other ground-based instruments. In addition, they are compared with satellite retrieved data sets. The current work also presents observed changes in total ozone and effective cloud transmittances.

## 3. Material and methods

### 3.1 Instruments in the Norwegian UV network

The GUV is a multi-wavelength filter radiometer manufactured by Biospherical Instruments Inc (BSI), San Diego (Bernhard et al., 2005). The detector unit is environmentally sealed and temperature stabilized, facilitating long-term reliable operation under harsh outdoor conditions. The GUVs have 5 channels in the UV range where each channel has a dedicated filter, a photodetector, and electronics that samples the output at a rate of about 3 Hz. The channels measure simultaneously global (direct and diffuse) solar irradiance at several UV wavelengths, which can be used to "reconstruct" the solar spectrum in the UV range and to compute biological doses, the UV Index, total ozone, and cloud transmittance.

The UV network consists of 12 multiband filter radiometers (model GUV-541 and GUV-511) (Bernhard et al. 2005). Nine of them are continuously operating at the network locations (Table 1) and three serve calibration purposes and are backups in case of failure at some of the stations. The instrument in Oslo/Kjeller is a GUV-511, whereas the instruments at the other sites are GUV-541. Both instrument types have four channels in the UV region (centre wavelengths 305, 320, 340, and 380 nm). In addition, GUV-541 has a fifth UV channel at 313 nm whereas GUV-511 has a fifth channel for measuring Photosynthetically Active Radiation (PAR: 400-700 nm). The bandwidths of the UV channels are ~10 nm (full width at half-maximum, FWHM). All instruments are temperature-stabilized at 40°C. Measurements are recorded as 1-minute averages, and for each instrument/site this represents several million records since the start in 1995.

84

The GUV-511 in Oslo was purchased already in 1993 and was installed at the University of Oslo (UiO) to test the instrument

performance and to develop appropriate software. In July 2019, this instrument was moved to Kjeller (~18 km East of UiO)

due to termination of total ozone/UV activity at the University of Oslo. Similarly, to assure continuation of the GUV time

series, the instrument in Tromsø was moved to the Arctic Lidar Observatory for Middle Atmosphere Research (ALOMAR)

facility at Andøya in 2000, about 130 km Southwest of Tromsø. Initial studies showed that the ozone climatology is very

similar at the two sites (Høiskar et al., 2001), however, the UV level is normally slightly higher at Andøya as the site is located

~50 km South of Tromsø.

With a few minor exceptions, the GUV instruments have been running continuously since 1995. The GUV at Andøya has been

subjected to some problems, most likely caused by an event of water intrusion. In spring 2013 an error with the 380 nm channel

was discovered and the instrument was sent to BSI for repair. Two years later, in 2015, the 320 nm channel failed and had to

be replaced. During these time periods spare GUV instruments were deployed from the DSA.

**Table 1: Overview of the locations, instrument types and institutes involved in the Norwegian UV network**

| Site | Location | GUV type (serial) | Supporting TOC instruments | Responsible institute |
|---|---|---|---|---|
| Landvik | 58.0°N, 08.5°E | GUV-541 | | DSA |
| Blindern, Oslo (1994-2019) | 59.9°N, 10.7°E | GUV-511 | Brewer#42 | NILU/UiO |
| Kjeller[1] (2019→) | 60.0°N, 11.0°E | GUV-511 | Brewer#42 | NILU |
| Østerås | 60.0°N, 10.6°E | GUV-541, GUVis-3511[2] | | DSA |
| Bergen | 60.4°N, 05.3°E | GUV-541 | | DSA |
| Finse | 60.6°N, 07.5°E | GUV-541 | | DSA |
| Kise | 60.8°N, 10.8°E | GUV-541 | | DSA |
| Trondheim | 63.4°N, 10.4°E | GUV-541 | | DSA |
| Andøya (2000 →) | 69.3°N, 16.0°E | GUV-541 | Brewer#104 | NILU[3] |
| Tromsø[4] (1995-1999) | 69.7°N, 17.0°E | GUV-541 | Brewer#104 | NILU |
| Ny-Ålesund | 78.9°N, 11.9°E | GUV-541 | Brewer#50[5,7], SAOZ, Pandora[6] | NILU[7] |

[1] GUV and Brewer#42 were moved from Blindern (University of Oslo) to Kjeller in June 2019

[2] GUVis-3511 was installed in 2018

[3] The instrument is inspected daily by staff at Alomar, Andøya Space Center

[4] GUV and Brewer#104 were moved from Tromsø to Andøya in the winter of 1999/2000

[5] Brewer#50 is operated by ISAC-CNR, Italy

[6] Pandora measurements started in the spring 2020

[7] The instrument is inspected daily by staff from the Norwegian Polar institute

As listed in Table 1 there are three Brewer spectrometers in operation in Norway: one in Oslo/Kjeller (B42), one in Tromsø/Andøya (B104), and one in Ny-Ålesund (B50). Generally, the Brewer instruments have been approved by the WMO as reliable high-quality instruments (WMO, 2018; Fioletov; 2008). The "Direct Sun" (DS) algorithm is the primary measurement mode of the Brewer and is based on measurements of the intensity of direct sunlight at five wavelengths between 306 nm and 320 nm. The precision of this method can be as high as 0.15% (Scarnato et al., 2010), but the absolute accuracy relies on an appropriate calibration. Under cloudy conditions, total ozone can be derived by measuring the intensity of scattered radiation from the zenith. As shown by Stamnes et al. (1990) there are some limitations of the zenith sky (ZS) method, but nevertheless this method provides useful information about total ozone content when the DS method cannot be used. Measurements of the Brewer global irradiance (GI) is an alternative to the ZS method and is also based on the principle of measuring scattered UV radiation from the sky.

The Norwegian Brewer instruments have been calibrated by the International Ozone Service (IOS, Canada) every year since installation in the 1990s, except from the summer 2020 when the calibration was prohibited under the COVID-19 restrictions. These frequent calibrations are done to ensure high quality Brewer measurements and to make sure that the instruments are well maintained and perform DS measurements within an accuracy of ±1%. The instrument B42 in Oslo is an MKV single-monochromator Brewer, which might be influenced by stray-light (Karppinen et al., 2015). Therefore, in this study we have only used Brewer DS data with ozone slant column below 1100 DU where the effect of stray light is negligible. All Brewer DS daily mean data from Oslo/Kjeller and Andøya are available at the World Ozone and Ultraviolet Radiation Data Centre (WOUDC, https://woudc.org/). Also, the Italian Brewer (B50) in Ny-Ålesund has been calibrated regularly by IOS Canada, last time in 2018, which showed that the instrument has been stable since the previous calibration in 2015. However, there are limited Brewer DS measurements available in Ny-Ålesund and the measuring season is relatively short due to the high latitude (79°N). Thus, in addition to Brewer DS data we have used SAOZ measurements to obtain quality assured ozone data from the early spring and fall. SAOZ derives total ozone from the Chappuis bands in the visible part of the spectrum through the Differential Optical Absorption Spectroscopy (DOAS) method (Pommereau and Goutail, 1988) and contrary to Brewer it can only measure ozone when the solar beam pathway through the atmosphere is large (solar zenith angle > 85°), i.e. around sunrise and sunset. Analyses and QC of the SAOZ data are performed at LATMOS (France) in the framework of the SAOZ global network (http://saoz.obs.uvsq.fr/index.html). In this study, we have used SAOZ daily average total ozone on days where both sunrise and sunset measurements are available. The data are stored in the Network for the Detection of Atmospheric Composition Change (NDACC) data base (http://www.ndaccdemo.org/). Based on experience and results from intercomparison campaigns, the SAOZ total ozone uncertainty is estimated to be within 3% (Hendrick et al., 2011).

The GUV data in the present study have been compared to OMI/Aura and GOME2/MetOp-A TM3DAM v4.1 total ozone data from Oslo, Andøya, and Ny-Ålesund. The satellite data from OMI and GOME2 are available from 2004 and 2007, respectively. These data are assimilated products, based on the TM3DAM software developed by Royal Netherlands Meteorological

Institute, KNMI (Eskes et al., 2003). The GOME2 and OMI assimilated TOC values are publicly available and are provided
on a daily basis via ESA's TEMIS project (http://www.temis.nl). The data files include error estimates, which are dependent
on location and time of year. During winter, the error can be as high as 8-10% whereas the error estimates usually are around
1-2% during summer.

In section 5.3, trends in effective cloud transmittance from the GUVs are discussed, and cloud data from the Norwegian Centre
for Climate Services (NCCS; https://klimaservicesenter.no) are being used to help in the interpretation of these measurements.
These data describe the number of clear-sky days observed every month. Cloud observations are performed three times per
day, but we have selected the measurements at 12:00 to reflect the period where GUV noontime values are measured. The data
describe the fraction of clouds as a number (NN) ranging from 0 to 8. NN=0 means clear sky, whereas NN=8 means completely
overcast. In our study we have classified the day as "clear" if NN at 12:00 has the value of 0, 1 or 2. NN=2 means that a quarter
of the sky is covered by clouds.

**3.2 Calibrations**
The procedure for calibrating the GUVs is described by Dahlback (1996) and only briefly presented below. When the GUV
Teflon diffuser is illuminated by a source, the photodetector transforms the radiation to an electric current which subsequently
is converted to a voltage signal. The measured voltage of channel $i$ is

$$V_i = k_i \int_0^\infty R_i'(\lambda)\, F(\lambda)d\lambda \approx k_i \sum_{\lambda=0}^{\infty} R_i'(\lambda)F(\lambda)\Delta\lambda \tag{1}$$


where $k_i$ is a constant (response factor), $R'_i(\lambda)$ is the relative spectral response function for channel $i$, and $F(\lambda)$ is the spectral
irradiance at wavelength $\lambda$. During the calibration, the Sun is used as the light source and the irradiance $F(\lambda)$ is measured by a
reference radiometer at the same time as the co-located GUV is recording the voltage $V_i$. The relative spectral response
functions for the GUVs were characterized at the optical laboratory of the Norwegian Radiation and Nuclear Safety Authority
(DSA). When $V_i$, $R'_i(\lambda)$, and $F(\lambda)$ are known, one can calculate the constant $k_i$ and the absolute response for channel $i$: $R_i(\lambda)=$
$k_i R'_i(\lambda)$.

The shape of the solar UV spectrum at the Earth's surface depends mostly on the solar zenith angle (SZA) and the TOC. Thus,
the spectral distribution of $R'_i(\lambda)F(\lambda)$ in Eq. (1) will depend on these parameters (Dahlback, 1996). The error in the derived
irradiance depends on how much the atmospheric conditions at the time of the measurement differ from those at the time of
the absolute calibration. This is discussed in more detail in Section 3.3.
The calibration procedure described above is normally done during large national or international intercomparison campaigns,
where the GUVs are operating synchronously with co-located high-resolution reference spectroradiometers. One of these
campaigns was arranged in Oslo in 2005, initiated through the national project "Factors Affecting UV Radiation in Norway"
(FARIN) (Johnsen et al., 2008; WMO 2008). Here the GUVs were intercompared with a Bentham spectroradiometer belonging
to DSA, which is closely linked to the Quality Assurance of Spectral Ultraviolet Measurements in Europe (QASUME) World
travelling reference spectroradiometer. Another large intercomparison campaign, which included the QASUME reference
spectroradiometer, was arranged in May/June 2019 (PMOD/WRC, 2019).

A key factor for the maintenance of a homogenous and stable calibration scale for the network instruments is a system for
quality control which accounts for long-term changes in the absolute response factors $k_i$. In Norway, this is implemented via a
dedicated travelling reference GUV-541, which has been transported to the respective stations every summer since 1995. The
traveling instrument is set up next to the stationary GUV and synchronous measurements with the two instruments are
performed for about one week. The irradiance from the two collocated instruments are compared to results from the 2005
calibration campaign, where the drift for all instruments and channels were set to unity. Relative to this 2005 calibration, yearly
drift factors $d_i$ for the individual channels (and instruments) are derived. These drift factors are used to modify the response
factor in Eq. (1), $k_i'=k_i/d_i$. If $d_i$ changes from one year to the next, a linear change in $d_i$ is assumed for periods between the two
intercomparisons. The method is described in more detail in WMO (2008). DSA is responsible for these annual assessments
of drift and determination of correction factors. Assessments of long-term drift $d_i^R$ for the travelling reference GUV itself are
made at the optical laboratory at DSA. Additionally, the travelling reference GUV has been shipped to the manufacturer every
one or two years since 1996 for an independent evaluation of long-term drift and for technical services.

**3.3 Retrievals of total ozone and effective cloud transmittance**
The GUV data products described in this work consist of measurements used in combination with a radiative transfer model
(RTM) based on the discrete ordinate method (Stamnes et al., 1988; Dahlback et al., 1991). When solar radiation passes through
the atmosphere, a portion of the UVB radiation will be absorbed by ozone. Other fractions of the radiation will be multiple
scattered or absorbed by air molecules, aerosols, and clouds (Stamnes et al., 2017). The total ozone column (TOC) is
determined from the GUVs by comparing a measured and calculated N-value, where the calculated N-value is derived from
the radiative transfer model. The N-value is defined as the ratio of irradiances in two different UV channels, with spectral
response functions $R_i(\lambda)$ and $R_j(\lambda)$. One of the channels is sensitive to total ozone whereas the other one is significantly less
sensitive. Hence, the N-value is defined as

$$N(SZA, TOC) = \frac{\sum_{\lambda=0}^{\infty} R_i(\lambda) F(\lambda, SZA, TOC) \Delta\lambda}{\sum_{\lambda=0}^{\infty} R_j(\lambda) F(\lambda, SZA, TOC) \Delta\lambda} = \frac{V_i}{V_j} \qquad (2)$$

where $F(\lambda, SZA, TOC)$ is the solar spectral irradiance at wavelength $\lambda$, solar zenith angle $SZA$, and TOC. $V_i$ and $V_j$ are the
measured voltages in channel $i$ and $j$, respectively. In this study the ratio channel (320nm)/channel(305nm) is used for
measuring and modelling the N values in Eq. (2). Prior to the measurements, the RTM has been used to create a lookup table
of N for all relevant combinations of SZA and TOC, and the GUV TOC is inferred by comparing the measured $V_i/V_j$ and
modelled N-values at the given SZA. The N-tables calculated from the RTM are for clear skies, but the table can also be
applied to cloudy skies because the effect of clouds on spectral irradiance at 305 nm and 320 nm is quite similar compared to
the large effect of ozone (Stamnes et al., 1991).

The N-tables described above are based on the 320/305 nm wavelength ratio and RTM calculations with the TOMS V7 ozone
climatology (McPeters et al., 1996), which describes an idealized altitude profile of temperature, pressure, and ozone. Previous
studies have shown that N-tables generated from this atmospheric profile agreed well with ozone values provided by the
Dobson spectrophotometer in Oslo during wintertime (Dahlback, 1996). Several other N-tables are created for the GUVs, both
for other wavelength ratios (e.g. 320/313 nm and 340/305 nm) and for subarctic summer and subarctic winter profiles (defined
by Anderson et al., 1987). The choice of ozone profile in the calculations of N-value lookup tables is especially important for
the winter when the SZA is large. Lapeta et al. (2000) found that an inappropriate ozone profile may cause uncertainties up to
10% in the retrieved TOC for SZA > 75°. Sensitivity studies from Dahlback (1996) showed that the errors in total ozone,
related to an inappropriate atmospheric profile in the RTM, was less than 1% for SZA < 65°. However, the error could be as
large as 30% (at SZA=80°) if a subarctic winter profile was replaced with a tropical atmospheric profile. This latter example
represents an extreme situation in Norway.

To quantify the effects of clouds, aerosols and changing surface albedo, a cloud transmission factor is introduced. It is defined
as the measured irradiance at wavelength channel $i$ and solar zenith angle SZA, $F_i(SZA)$, compared to the modelled irradiance
at a cloudless and aerosol-free sky with a none-reflecting surface, $F_{ic}(SZA)$. $F_{ic}$ is calculated for the same wavelength and solar
zenith angle as the actual measurement $F_i$, and a channel insensitive to ozone absorption is selected. The estimates of cloud
transmission and optical depth are sensitive to ground reflection, implying that an accurate determination of cloud attenuation
requires precise knowledge of the surface albedo. Stamnes et al. (1991) introduced the term *effective* cloud transmittance to
account for the influence of surface albedo and aerosols on cloud attenuation. The effective cloud transmittance (eCLT) is
defined as:

$$eCLT(SZA) = 100 \frac{F_i(SZA)}{F_{ic}(SZA)} \qquad (3)$$


In this study, the 340 nm channel has been selected to determine the eCLT. Alternatively, the 380 nm channel can be used, but
the derived eCLT is virtually independent of the choice of the 340 or 380 nm channel. For both wavelengths, the incoming
solar radiation is insensitive to ozone, meaning that the eCLT is only sensitive to clouds, aerosols, and the surface albedo.
ECLT may be larger than 100% due to multiple scattering of the solar radiation when broken clouds are present and the sun
remains unobscured. Furthermore, the presence of snow on the ground enhances the albedo and contributes to an additional
multiple scattering. We do not attempt to separate the effects of clouds, aerosols, and albedo here, and the eCLT quantifies the
combined influence of the three factors.

**4. Data analysis**
**4.1 Harmonization of total ozone**
As described in Section 3.3, each GUV instrument has a unique set of N-tables, and to obtain optimal ozone measurements it
is possible to switch between various tables depending on season and solar zenith angle. However, in our study we have only
used one N-table for a given station (with TOMS V7 ozone climatology (McPeters et al., 1996) and 320/305 nm channel ratio)
to simplify the ozone estimates and avoid artifacts in trends and statistics generated by lookup table (N-table) changes. To
account for possible seasonal errors in total ozone related to the above-mentioned inaccuracies in the atmospheric profile and
variations in surface albedo (snow/ice on the ground), we have homogenized the GUV measurements with respect to Brewer
Direct Sun (DS) total ozone measurements. All Brewer DS data are daily mean values, identical to the data available at the
WOUDC data base.

Figure 2 shows the GUV/Brewer DS ratio for the period 1995-2018 for days with available GUV and Brewer DS (and SAOZ)
data. The GUV daily average total ozone values are calculated as 1h averages around local noon, and to limit possible errors
caused by clouds, we have selected days where the noontime average eCLT from GUV is larger than 60%. Also, GUV
noontime TOC with standard deviation larger than 20 DU have been flagged as "uncertain" and are not included in the data
analysis. Comparisons between GUV (global sky) and Brewer DS time series in Figure 2 demonstrate highly consistent results,
i.e. the individual instruments have been stable and homogenous since the start in 1995.

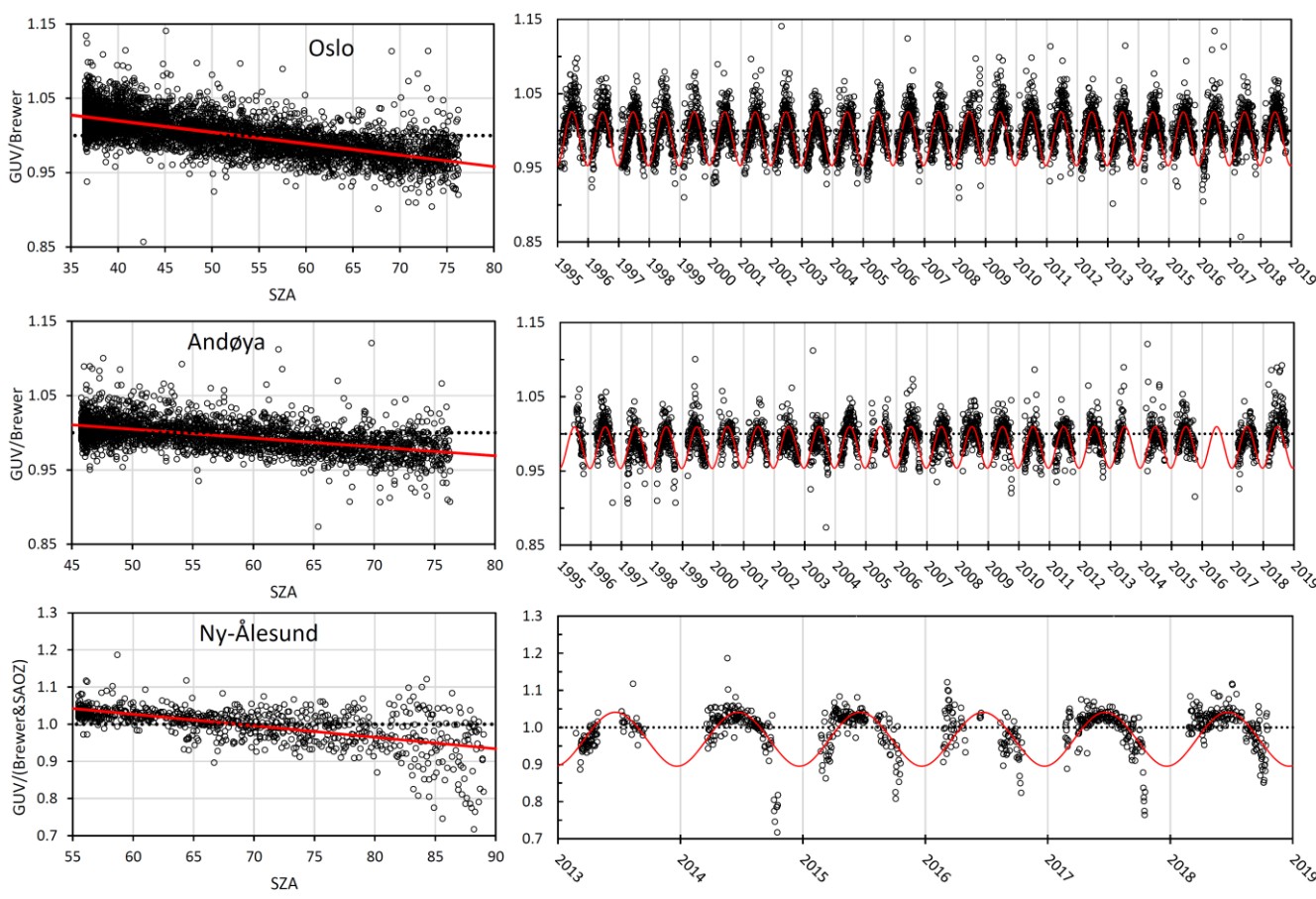


**Figure 2: Ratios of GUV/Brewer(DS) ozone values measured in Oslo (top), Tromsø/Andøya (center) and in Ny-Ålesund (bottom). SAOZ ozone data are also used in Ny-Ålesund. The left panels show TOC ratios as a function of SZA, where the red lines represent the linear fit. The right panels show daily TOC ratios for all years with simultaneous measurements. The statistical fit functions are marked as red curves.**

As seen from Figure 2 there is a clear seasonality in the TOC ratio. This can both be attributed to an instrumental SZA dependence and/or a seasonal variability related to the atmospheric profile in the RTM and N-tables used for ozone retrievals. Inspections of GUV minute values performed throughout a day do not necessarily give a very clear explanation of the variability. Figure 3 shows two examples from April and June 2018, where GUV TOCs in Ny-Ålesund, normalized to noontime TOC (TOC_noon), are plotted throughout the day. The plot from April (Figure 3, top panel) does not indicate any obvious SZA dependence in the measurements. However, there is a significant spread in the ratio as SZA exceeds 82°, mainly due to noise in measurements of the 305 nm channel. This might mask a possible SZA dependence. Also, spring-time ozone has normally large day-to day variations and the morning TOC will often differ from the evening value. Contrary to the upper

panel, the bottom panel in Figure 3 (from June 2018) indicates a clear decrease in TOCs as SZA increases. At SZA=78°, which
is the maximum SZA at midnight in Ny-Ålesund in June, the average ratio TOC/TOC_noon is 0.97. For calculations of the
harmonized noon-time TOC it is of minor importance whether the ozone values are corrected from a SZA or "day-of-year"
statistical fit function, but based on inspections of a number of daily minute values (such as Figure 3, lower panel) a SZA
correction is considered to give the best physical interpretation of the annual TOC variability.

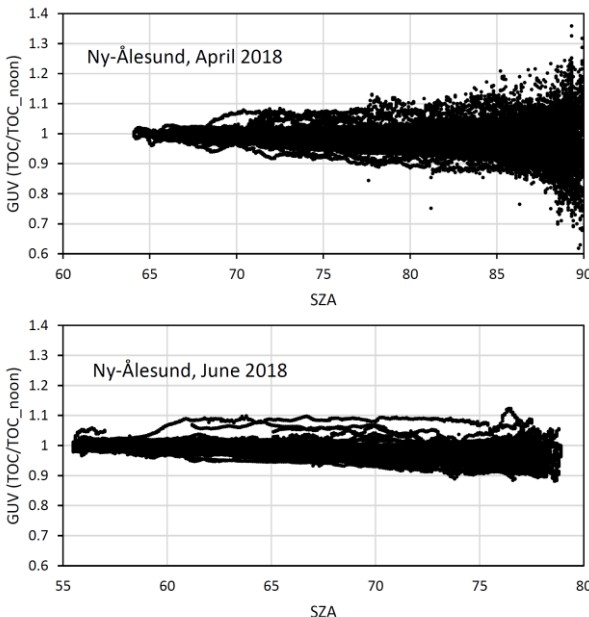


**Figure 3: GUV TOC from Ny-Ålesund measured throughout two selected periods: April 2018 (upper panel) and June 2018 (lower**
**panel).**

When all measurements and seasons are considered as a whole, we have chosen an SZA correction of GUV TOC data to
harmonize with other ground-based instruments at the stations. All available GUV/Brewer DS (and SOAZ) ratios have been
fitted by the linear functions $f(SZA)$ indicated by a red line in the left panels of Figure 2:

$$f(SZA) = a * SZA + b \qquad\qquad (4)$$


Here $a$ and $b$ are constants listed in Table 2 for the individual stations. The SZA corrected total ozone value (TOC') is computed
as TOC'=TOC/$f(SZA)$.

 **Table 2: Results from statistical fit of GUV/Brewer(& SAOZ) ratio, *a* is the slope and *b* is the constant in Eq. (4). *S*tandard deviation**
**(*STD*) of the coefficients are included.**

| Station | $a \pm STD$ | $b \pm STD$ |
|---|---|---|
| Oslo | -0.00154 ± 4E-05 | 1.0814 ± 0.0018 |
| Andøya/Tromsø | -0.00119 ± 4E-05 | 1.0642 ± 0.0025 |
| Ny-Ålesund | -0.0031 ± 2E-04 | 1.2129 ± 0.0112 |



The harmonization method described above are applied to the three GUVs operated by NILU, which are co-located with other
ground-based ozone monitoring instruments. Total ozone is also derived for the other stations in the UV network (presented
in Table 1 and Figure 1), but for these instruments a different approach is used. A description of the method and results will
be presented in a separate paper.

**4.2 Ozone cloud correction**
Under heavy cloud conditions the ozone retrievals are usually less accurate. An extreme example is discussed by Mayer et al.
(1998) for a thunderstorm case. They found that multiple scattering caused errors as large as 300 DU. A less extreme situation,
which is more representative for Norway, is exemplified in Figure 4. The figure shows eCLT (black line) and total ozone
column (red line) derived from GUV measurements in Oslo between 11:00 and 17:00 UTC on 9 September 2018. Figure 4
indicates a gradual ozone decrease throughout the day, but what is most interesting is the occurrence of ozone peaks when
eCLT is very low. The uncertainty in total ozone increases as the cloud optical depth becomes very large, and normally we
use a cutoff at eCLT=20% and do not accept ozone retrievals under these heavy cloud conditions.

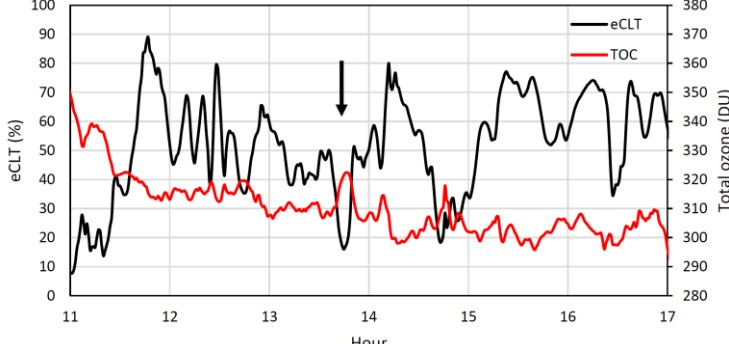


**Figure 4: Total ozone and eCLT during a day (9 September 2018) with heavy clouds at Blindern, University of Oslo. Black arrow**
**indicates a time where eCLT drops below 20%.**

The example in Figure 4 shows that total ozone increases by 15 DU (~5%) when eCLT drops from 50% to 16% (see arrow in Figure 4). However, the eCLT effect on ozone is less evident for thinner clouds. In order to examine the impact of clouds on TOC more systematically, we analyzed the difference between SZA corrected GUV noontime TOC and Brewer DS (& SAOZ) values as a function of eCLT, using data starting in 1995. Brewer DS measurements are not performed during cloudy conditions, so these measurements are typically done during a "clear" period on the same day as GUV recorded clouds around noon. The results for Oslo, Andøya, and Ny-Ålesund are shown in Figure 5 for observations with SZA < 80°. The figure shows that the ozone ratios are characterized by gradual decreases for eCLT ranging between 20% and 60%, while for eCLT > 60% the ratios vary around one.

321

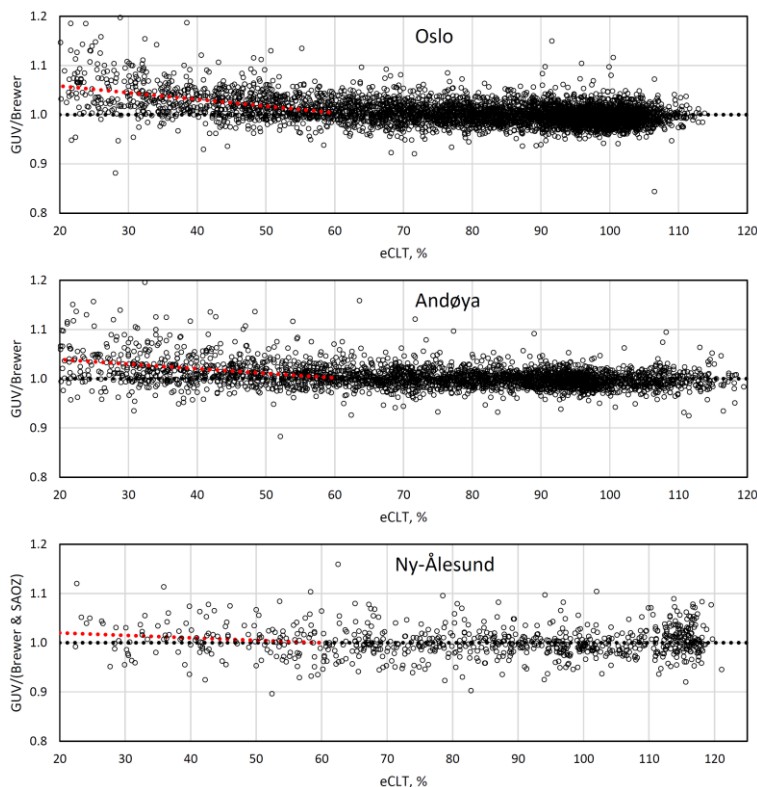

**Figure 5: Ozone difference between GUV and Brewer DS (& SAOZ) as a function of eCLT: Oslo (top), Andøya (center) and Ny-Ålesund (bottom). The red dotted lines indicate the linear best fitting for eCLT < 60%. The presence of eCLT higher than 100% is discussed in Section 3.3.**

Based on this analysis we have introduced a linear ozone correction *g(eCLT)* for eCLT< 60%,

326

$$g(eCLT) = \alpha * eCLT + \beta \qquad\qquad (5)$$

where $\alpha$ represents the slope and $\beta$ is a constant. The values of $\alpha$ and $\beta$ for Oslo, Andøya, and Ny-Ålesund are summarized in Table 3. For Ny-Ålesund there are few Brewer DS and SAOZ data available on days with heavy clouds, and consequently the eCLT correction function is more uncertain than the one for Oslo and Andøya. This is also reflected from the high standard deviation of $\alpha$ in Table 3. The overall eCLT correction for Ny-Ålesund is relatively small, i.e. a 2% correction when eCLT drops from 100% to 20%. The corresponding ozone corrections for Oslo and Andøya are ~5% and ~4%, respectively.

**Table 3: Ozone cloud correction for eCLT < 60%, where $\alpha$ is the slope and $\beta$ is the constant in Eq. (5). Standard deviation (*STD*) of the coefficients are included.**

| Station | $\alpha \pm STD$ | $\beta \pm STD$ |
|---|---|---|
| Oslo | -0.00137 ± 0.00011 | 1.0822 ± 0.0050 |
| Andøya/Tromsø | -0.00093 ± 0.00015 | 1.0558 ± 0.0068 |
| Ny-Ålesund | -0.00050 ± 0.00040 | 1.0300 ± 0.0185 |

The full GUV TOC time series from 1995 and onwards have been harmonized with respect to the SZA and eCLT corrections described above. Specifically, TOCs have been divided by the fit-function *f(SZA)* in Eq. (4). For cloudy conditions with effective cloud transmittance less than 60% an additional correction *g(eCLT)*, given in Eq. (5), has been applied to the data. With this harmonization, accurate GUV total ozone values can be retrieved under most conditions. Table 4 gives an overview of correlation, bias and standard deviation between GUV and Brewer DS (& SAOZ) for the original GUV data sets, shown in Figure 2, and for the final corrected data sets. As expected, the correlation increases and the standard deviation (STD) is reduced after the GUV harmonization. The biases for the final data sets are all within ±0.3%. The STD of the GUV-Brewer (& SAOZ) difference is 2.5%, 2.4%, and 4.5% for the Oslo, Andøya, and Ny-Ålesund time series, respectively. This is a reduction of 0.5-1.1% compared to STD for the uncorrected data sets.

**Table 4: Correlation, bias, and STD in total ozone from GUV and Brewer (& SAOZ). The left columns are for uncorrected GUV data, whereas the right columns are for SZA and CLT corrected GUV total ozone data. Bias and STD are both expressed in DU and % (in parenthesis)**

| Station | Uncorrected | | | Corrected | | |
|---|---|---|---|---|---|---|
| | Correlation | Bias, DU (%) | STD, DU (%) | Correlation | Bias, DU (%) | STD, DU (%) |
| Oslo | 0.969 | 2.9 (0.9) | 12.0 (3.6) | 0.984 | -0.1 (0.0) | 8.5 (2.5) |
| Andøya | 0.983 | 0.1 (0.0) | 9.9 (2.9) | 0.989 | -0.3 (-0.1) | 8.4 (2.4) |
| Ny-Ålesund | 0.966 | 0.8 (0.2) | 17.8 (5.1) | 0.976 | 0.9 (0.3) | 15.7 (4.5) |

350

The ratios between GUV and Brewer DS (& SAOZ) TOC are visualized in Figure 6 for the three stations; Oslo (top), Andøya (center) and Ny-Ålesund (bottom). Compared to Figure 2 no systematic seasonality can be seen in the ratios. Ny-Ålesund is possibly an exception, where low GUV TOC values are seen in late fall most of the years. These measurements are performed at very high SZA (84-89°) where the GUV uncertainty is high. If we only consider GUV measurements with SZA<82° the high/low ratios in fall and spring disappears and the standard deviation between GUV and Brewer (& SAOZ) is reduced to 3.5%.

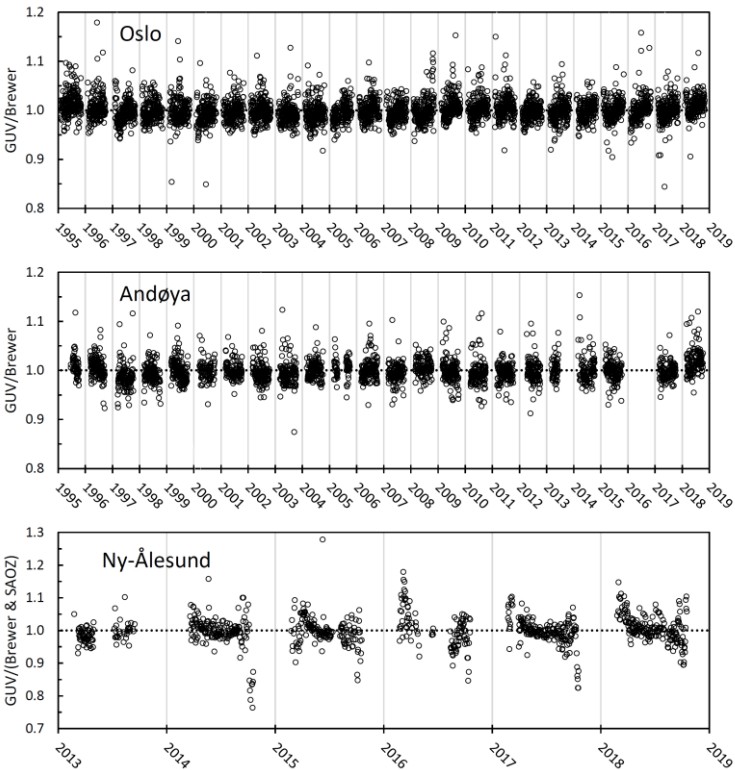

**Figure 6: Ratios of GUV/(Brewer & SAOZ) ozone values measured in Oslo (top), Tromsø/Andøya (center), and Ny-Ålesund (bottom) for the GUV corrected data sets. Measurements for all SZA and eCLT values are included.**

## 5. Results

### 5.1 Comparison with total ozone column from satellites

Corrected GUV TOCs have been compared to GOME2-A and OMI TM3DAM v4.1 (Eskes et al., 2003) data for Oslo, Andøya, and Ny-Ålesund. It should be emphasized that GUV data are homogenized with respect to Brewer DS (and SAOZ) data and that any offset between Brewer and satellite data most likely will be reflected by offset in GUV-GOME2 and GUV-OMI ozone data. Figure 7 shows the difference (in %) of daily noontime GUV and GOME2 total ozone for the period 2007-2019 (left column) and GUV vs OMI for the period 2004-2019 (right column). Results for Oslo are shown in the top row, Andøya in the center row and Ny-Ålesund in the bottom row. The correlations, biases and STDs are listed in Table 5. At Oslo, the noontime total ozone is never calculated at SZA> 83°, which is the noontime SZA at the winter solstice. As seen from the figure, the spread in the GUV-GOME2 difference increases as SZA exceeds 82°, especially for Andøya. The statistics presented in Table 5 also indicates that the overall STD for Andøya is larger than for the other locations. The reason for this is not entirely clear but can partially be attributed to a combination of uncertainties in GUV and satellite measurements at this coastal area where clouds, albedo, and topography vary on a small scale. For example, drifting clouds at Andøya occur frequently and lead to a large variability in the ratio of satellite and ground-based UVI measurements during spring and summer when albedo is low (Bernhard et al. 2013). Further, clouds represent an atmospheric factor that can significantly reduce the accuracy of both ground-based measurements and satellite TOC data (Antón and Loyola, 2011).

379

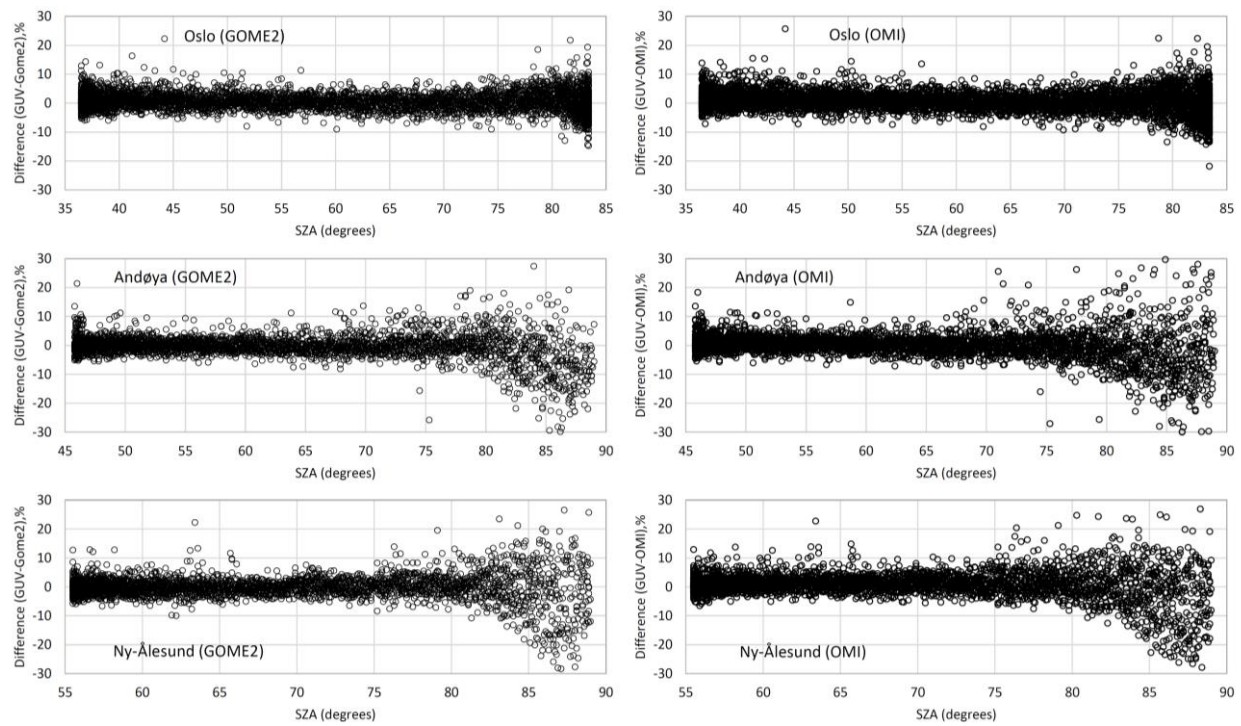

**Figure 7: Total ozone differences (in %) between GUV and GOME2 (left column) and GUV-OMI (right column): Oslo(top), Andøya**
**(center), and Ny-Ålesund (bottom)**

**Table 5: Correlation, bias, and STD in daily noontime total ozone from (a) GUV vs GOME2 2007-2019, and (b) GUV vs OMI 2004-**
**2019. Bias and STD are both expressed in DU and % (in parenthesis)**

| | (a) GUV vs GOME2 | | | | | |
|---|---|---|---|---|---|---|
| | **All SZA** | | | **SZA<80°** | | |
| **Station** | **Correlation** | **Bias, DU (%)** | **STD, DU (%)** | **Correlation** | **Bias, DU (%)** | **STD, DU (%)** |
| Oslo | 0.974 | 2.2 (0.6) | 11.2 (3.4) | 0.979 | 2.4 (0.7) | 10.1 (3.0) |
| Andøya | 0.954 | -1.3 (-0.4) | 19.7 (5.8) | 0.983 | 1.0 (0.3) | 11.0 (3.3) |
| Ny-Ålesund | 0.966 | 0.2 (0.1) | 17.7 (5.1) | 0.986 | 0.0 (0.0) | 9.6 (2.8) |
| | (b) GUV vs OMI | | | | | |
| | **All SZA** | | | **SZA<80°** | | |
| **Station** | **Correlation** | **Bias, DU (%)** | **STD, DU (%)** | **Correlation** | **Bias, DU (%)** | **STD, DU (%)** |
| Oslo | 0.968 | 1.7 (0.5) | 12.9 (3.9) | 0.977 | 2.8 (0.8) | 10.8 (3.2) |
| Andøya | 0.904 | 2.0 (0.6) | 28.9 (8.6) | 0.972 | 2.2 (0.6) | 14.0 (4.1) |
| Ny-Ålesund | 0.963 | 2.8 (0.8) | 18.2 (5.3) | 0.984 | 3.8 (1.1) | 10.5 (3.1) |

Figure 7 and Table 5 show that GOME2 gives slightly better agreement with GUV TOC compared to OMI. For all stations, the STD is higher for GUV-OMI than for GUV-GOME2, both when the entire GUV time series and data with SZA < 80° are considered. The standard deviations of the GUV-GOME2 differences range from 3-6% when all measurements are included but is reduced to ~3% if we only consider measurements with SZA < 80°. For GUV-OMI the corresponding STDs are in the range 4-9% if all measurements are included and 3-4% if data with SZA < 80° are used. The overall biases between GUV and satellite data are within ±1% for all stations, but on average, OMI is slightly lower than GOME2, especially at the two northernmost stations.

## 5.2 Long-term changes in total ozone

For total ozone assessment and trends studies, the established Brewer instruments would normally be used. However, as demonstrated in previous sections, GUV measurements can provide realistic and stable time series and are suitable for separate studies of long-term changes of the ozone layer. GUVs that are co-located with a Brewer or another standard TOC instrument for 2-3 years (until harmonization parameters are established), can afterwards be moved to a new location for independent TOC measurements. The harmonization procedure is used to minimize small systematic errors in GUV TOC data and assumes that Brewer data are without error. However, it should be noted that TOC retrievals at large SZAs can be uncertain if the new site has a very different ozone climatology compared to the original site, as explained in section 3.3. Data from the GUV instruments are also very useful to extend the measuring season at sites with reduced staff and/or characterized by harsh environmental conditions. The case of Ny-Ålesund, where Brewer data are very sparse due to a rough climate that require a high attendance, is a clear example of GUV usefulness. In Ny-Ålesund as much as 52% of TOC daily means have solely been based on GUV measurements during the last five years.

Even at sites like Oslo and Andøya, where good attendance and less harsh conditions allow more robust Brewer operations, GUV TOC can fill in missing data and extend the measuring season. Brewer zenith sky (ZS) or global Irradiance (GI) measurements (WOUDC 2019) are normally performed under cloudy conditions. However, these measurements can also be impacted by high SZA, heavy clouds or technical problems. The last five years, 14% of the daily mean TOC values at Andøya are retrieved from GUV to fill in for missing Brewer DS/ZS/GI measurements.

The overall GUV data coverage at the Norwegian stations is very good. If we disregard the two calibration campaigns in 2005 and 2019, the GUV-511 in Oslo has been in operation ~99% of all days since the start in 1995. Missing days are mainly caused by power failure or minor technical computer issues. TOC retrievals are performed ~95% of all days, where the missing retrievals usually are related to heavy cloud conditions (eCLT<20%) with high uncertainty. Due to the long and continuous

GUV time series, trend analyses based on these data will give a very good picture of the development of the ozone layer above Norway after 1995, along a very wide latitudinal range.

The GUV network was established during a period where a significant downward trend in total ozone had been observed for most places on Earth. Statistical analysis of the Dobson (D56) time series from Oslo 1978-1998 revealed an annual average total ozone decrease of -5.2 ± 0.6 %/decade during this period (Svendby and Dahlback, 2002). For the Norwegian stations, a minimum in annual average total ozone was measured during the period 1993-1997 (Svendby et al., 2020). Thus, a study of trend in GUV total ozone should also consider a possible influence by the low values the first few years.

Linear trends in the annual average total ozone at the three stations have been calculated, and the results are shown in Figure 8: Oslo in the top panel, Andøya in the center panel and Ny-Ålesund in the bottom panel. For the Oslo station we have a full year of data in 1995, whereas the measurements in Tromsø (Andøya) and Ny-Ålesund started in mid-1995 and a full year of data is not available until 1996. Thus 1995 is omitted from the time series at these two stations. Results from the linear regression analyses are presented in Table 6. In addition to changes in annual mean total ozone, the table includes also linear trends for winter (Dec-Feb), spring (Mar-May), summer (Jun-Aug), and fall (Sep-Nov).

The annual means in Oslo are based on data from January to December, for Andøya the means are calculated for the months from February to mid-November, whereas data from Ny-Ålesund are based on data from March to October. For the two northernmost stations the winter averages cannot be retrieved because of the polar night. Note also that the fall trend results for Ny-Ålesund, presented in Table 6, do not include November.

Due to different months included in the Oslo, Andøya and Ny-Ålesund annual means, the absolute values are not comparable. Still, there are many similarities in the three data sets. Even though Oslo and Ny-Ålesund are separated by more than 2000 km, the years with low annual average TOC often coincide. Annual variations in the ozone transport from its source region in the tropics toward the polar regions during the winter, will often have similar impacts at all our stations, and variations in Quasi-Biennial Oscillation (QBO), El Nino-Southern Oscillation (ENSO), the solar cycle, and stratospheric aerosols will give significant interannual variability in total ozone (WMO 2018; Svendby and Dahlback, 2004). The explanatory variables mentioned above are often used in TOC trend studies to eliminate variability caused by natural sources and to get a more precise picture of trends related to emissions of anthropogenic sources such as ODSs.

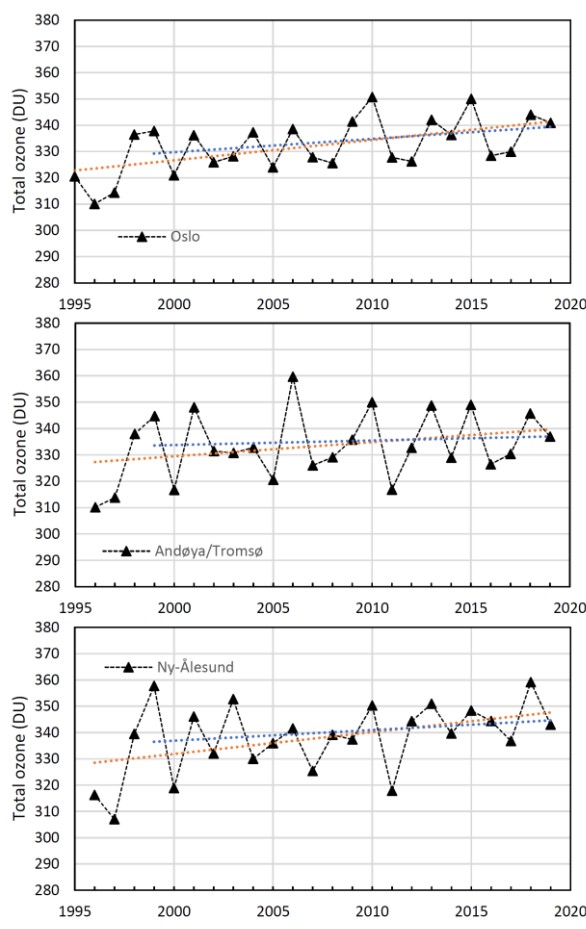

**Figure 8: Annual average total ozone in Oslo, at Andøya/Tromsø, and in Ny-Ålesund. Linear trends for the whole period 1995(96)-2019 are marked with orange lines, ozone changes for 1999-2019 are in blue.**

In Figure 8, linear observational trends for the entire period (from 1995(96) to 2019) are marked in orange, whereas changes for the last 20 years are marked in blue. The latter trend estimate is done to eliminate the years in the mid-1990s with very low ozone, partly influenced by the Mt. Pinatubo eruption and the cold Arctic winters in 1996 and 1997 (Solomon et al., 1999). The analysis reveals a total ozone increase for the period 1995(96)-2019 at all stations and for all seasons. However, only half of the positive trend results are statistically significant to a 95% confidence level (2σ), that is annual trends in Oslo (2.3 ± 1.5%/decade) and Ny-Ålesund (2.5 ± 2.5%/decade), fall trend in Oslo (3.4 ± 1.5%/decade) and Andøya (3.0 ± 2.8%/decade), and spring values in Ny-Ålesund (3.8 ± 3.5%/decade). If we exclude the years 1995-1998 and only look at the changes for the period 1999-2019, the regression analysis still indicates an increase in total ozone during the last two decades. However, the increases are less pronounced and not significant at the 2σ level, except from the increase in Oslo (3.2 ± 2.0%/decade) for fall.

The annual TOC trends for the 1999-2019 period are 1.5 ± 1.8 %/decade for Oslo, 0.5 ± 2.6 %/decade for Andøya, and 1.2 ± 2.4%/decade for Ny-Ålesund. Results that are statistically significant are marked in bold in Table 6. Total ozone is strongly influenced by stratospheric circulation and meteorology, which give rise to large interannual variability in total ozone. This variability will reduce the statistical significance and can mask a potential trend in total ozone. The overall positive trend results from the three Norwegian stations agree well with analyses from the "Scientific Assessment of Ozone Depletion: 2018" (WMO 2018). Model simulations presented in WMO (2018) conclude that about half of the observed upper stratospheric ozone increase after 2000 is attributed to the decline of ODSs since the late 1990s. The other half of the ozone increase is attributed to the slowing of gas-phase ozone destruction cycles, which results from cooling of the upper stratosphere caused by increasing concentrations of greenhouse gases. It should be noted that stratospheric cooling reduces Arctic ozone if the temperature drops below the threshold of formation of polar stratospheric clouds (PSCs), as exemplified below. Normally PSCs will only exist between December and March and therefore mainly affect ozone trends for winter and early spring.

**Table 6: Seasonal and annual changes in total ozone in Oslo, at Andøya and Ny-Ålesund for the period (a) 1995 – 2019 (start year 1996 for Andøya and Ny-Ålesund), (b) 1999-2019. Uncertainty is expressed as 2*STD (2σ).**

| (a) TOC observational change, %/decade 1995(96)-2019 | | | | | |
|---|---|---|---|---|---|
| | winter | spring | Summer | Fall | Annual |
| Oslo | 2.92 ± 3.23 | 1.68 ± 2.27 | 0.97 ± 1.27 | **3.38 ± 1.50** | **2.33 ± 1.46** |
| Andøya | | 1.30 ± 2.59 | 0.77 ± 1.37 | **2.95 ± 2.82** | 1.62 ± 2.22 |
| Ny-Ålesund | | **3.84 ± 3.45** | 0.96 ± 1.28 | 2.02 ± 4.50 | **2.46 ± 2.15** |
| (b) TOC observational change, %/decade 1999-2019 | | | | | |
| | winter | spring | Summer | Fall | Annual |
| Oslo | 1.75 ± 4.01 | 0.61 ± 2.85 | 0.68 ± 1.04 | **3.23 ± 2.01** | 1.54 ± 1.79 |
| Andøya | | -0.39 ± 2.76 | 0.88 ± 1.50 | 3.00 ± 3.69 | 0.51 ± 2.60 |
| Ny-Ålesund | | 1.39 ± 3.55 | 0.80 ± 1.56 | 1.52 ± 5.76 | 1.21 ± 2.42 |

Despite a general increase in TOC during the last decades, Lawrence et al. (2020) reported that the TOC over the northern polar region was exceptionally low in late winter and early spring 2020. The average total ozone for February to April was the lowest value registered since the start of satellite measurements in 1979. The low TOC was partly caused by an exceptionally cold and persistent stratospheric polar vortex, which provided ideal conditions for chemical ozone destruction (Grooß and Müller, 2020; Manney et al., 2020; Wohltmann et al., 2020). These low ozone values resulted in enhanced UV-radiation, and the average UV index measured by the GUV in Ny-Ålesund in April 2020 was elevated by 34% relative to the average 1979–2019 level (Bernhard et al., 2020).

Figure 9 shows GUV total ozone in Ny-Ålesund from mid-February to May 2020, and the low ozone levels from the end of
March to mid-April are clearly seen. Total ozone from SAOZ, GOME2 and OMI (TM3DAM v4.1) are included in the figure
for comparison. The study from Wohltmann et al. (2020) showed that the Arctic ozone at 18 km altitude was depleted by up
to ~93% in the spring 2020, which is comparable to typical local values in the Antarctic ozone hole. The agreement between
GUV, GOME2 and OMI is good during this ozone loss period, indicating that GUV performs well even though the ozone
profile used in the look-up table did not match the actual profile that was observed above Ny-Ålesund in March and April
2020. Figure 9 shows that the ground-based instruments, both GUV and SAOZ, in general give higher TOC than the satellites
during February and parts of March 2020. There is also a notable difference between GOME2 and OMI between mid-April
and the end of May. The satellite error estimates are around 4% for these months, and as explained in Section 3 the ground-
based instruments have also a significant uncertainty at SZA > 80°. This demonstrates the challenges of performing accurate
TOC measurements in the Arctic.

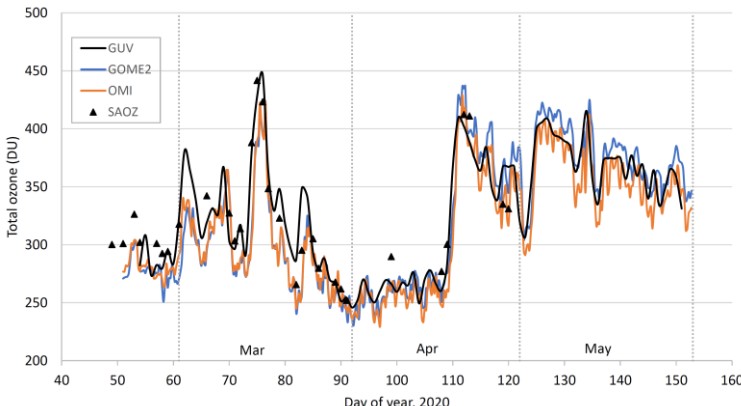



**Figure 9: Total ozone column measured in Ny-Ålesund the spring 2020 with the SAOZ instrument (black triangles), GUV (black**
**line), OMI satellite (orange line) and GOME2 (blue line).**

Episodes of very low total ozone content are not limited to early spring and periods of several weeks. They can also occur for
a few days because of unusual meteorological or atmospheric conditions, as observed at Kjeller in late 2019. In Figure 10,
GUV noontime total ozone from Oslo and Kjeller in 2019 is compared to GOME2 and OMI data from Oslo (12:00 values).
The black line shows GUV TOC data, whereas blue and orange lines represent GOME2 and OMI measurements, respectively.
The lack of GUV data from mid-May and June is caused by the calibration campaign at DSA (see section 3.2). GUV data prior
to mid-May 2019 are from Oslo, whereas measurements after July 2019 were performed at Kjeller outside Oslo. GUV
comparison to GOME2 and OMI overpass data from Oslo indicates that the agreement between ground-based measurements
and satellite data are as good at Kjeller as in Oslo. A very interesting episode is the extremely low total ozone values measured
on 4 December 2019 (red circle in Figure 10). On this day, the noontime GUV ozone value at Kjeller was only 193 DU. This
is the lowest value measured by the GUV in Oslo/Kjeller the last 20 years. GOME2 and OMI from Oslo also measured very
low total ozone at 12:00 this day, 201 DU and 203 DU, respectively. At 18:00 the previous day the total ozone value from
OMI was as low as 193.5 DU.

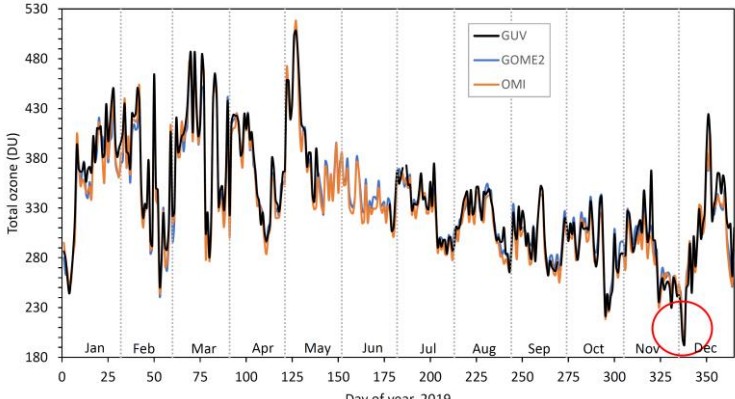


**Figure 10: Total ozone column values from Oslo/Kjeller in 2019 measured with the GUV instrument (black line), OMI satellite**
**(orange line) and GOME2 (blue line). The red circle indicates the mini "ozone hole" over Scandinavia 4 December 2019.**

In the fall/winter 2019 the Arctic polar vortex formed earlier than usual (Manney et al., 2020, Lawrence et al., 2020).
Temperatures were low enough for PSC formation by mid-November 2019, earlier than in any previous year since at least
2004. PSCs were visible over Norway during a large part of the winter 2019/2020. However, in early December, chorine
activation and associated chemical ozone loss was still limited. Dameris et al. (2021) indicate that a "mini ozone hole" over
Southern Norway on 4 December 2019 was caused by advection of lower-latitude airmasses and increased tropopause height.
Figure 11 shows total ozone from the GOME2 satellite at 12:00 this day. As seen from the figure the TOC was below 200 DU
in the middle parts of Norway, Northern Sweden, and South-Western Finland.

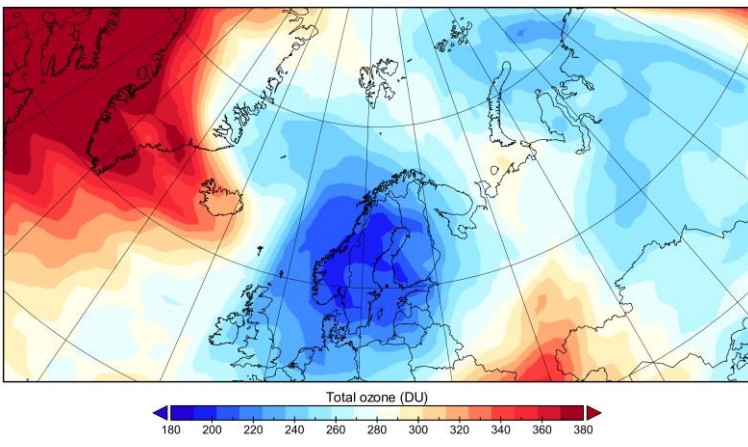

**Figure 11: Total ozone column on 4 December 2019 at 12:00 from the GOME2-A satellite (data downloaded from http://www.temis.nl/protocols/o3field/o3field_msr2.php)**


### 5.3 Trends in eCLT

As described in Section 3, the effective cloud transmittance (eCLT) expresses the effect of clouds, aerosols and surface albedo on the UV radiation reaching the ground. In the present study an eCLT of 100% represents a clear sky with no surface reflection. An eCLT value above 100% can occur in case of scattered clouds and/or enhanced surface reflection, e.g. snow.

Figure 12 shows annual average noontime eCLT values and trends at the three stations: Oslo (orange line), Andøya/Tromsø (grey/black line) and Ny-Ålesund (blue line). Linear regression analyses indicate that there are no changes in eCLT at Oslo or Andøya. However at Ny-Ålesund, eCLT has decreased over the last 25 years and a negative trend of 5-6% is evident from Figure 12. The change in eCLT is even more pronounced if we only consider the months from late spring and early summer (Apr-Jun), as shown in Figure 13. For these three months the overall decreases in eCLT are ~15% for April and May and 9% for June. The decadal trend is -7.6, -7.2, and -3.6 % for April, May, and June, respectively (Table 7).


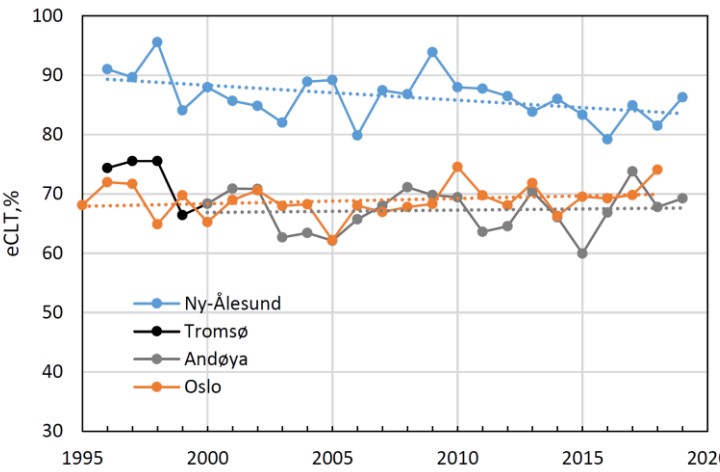


**Figure 12: Annual average noontime eCLT measured in Oslo, Tromsø/Andøya, and in Ny-Ålesund from 1995(96) to 2019. Trends in eCLT are indicated as dotted lines.**


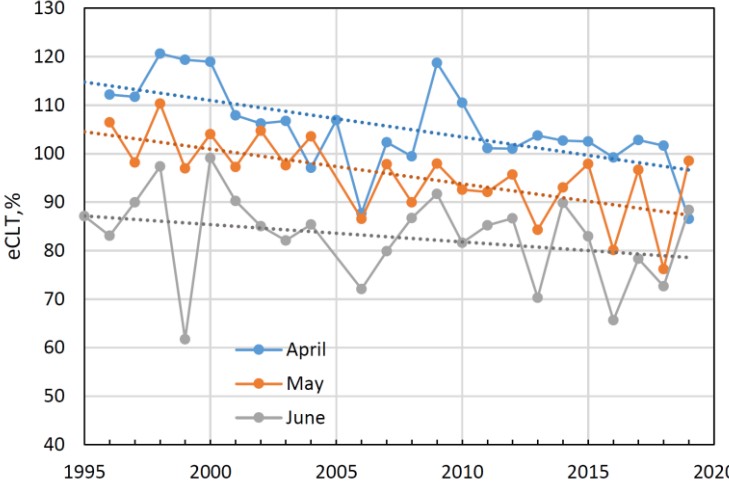


**Figure 13: Monthly mean eCLT in Ny-Ålesund for April, May, and June 1995(96) to 2019. Trends in eCLT are indicated as dotted lines.**


To examine possible monthly differences and changes in the cloud cover in Ny-Ålesund for the period 1995-2019, cloud data from the Norwegian Centre for Climate Services (NCCS) has been utilized (see section 3.1). NCCS cloud data at 12:00 have been selected to reflect the period where GUV eCLT noontime values are measured. Figure 14 shows the number clear days for April (blue), May (orange) and June (black) for the years 1995-2019. The average number is ~10 days for April, ~7 days for May, and only ~4 days for June. Naturally, there are some variations from one year to another, but for the period 1995-

2019 it is an overall decrease in the number of clear-sky days. The dotted lines in Figure 14 indicate that there has been an average monthly decrease of 2-3 clear days during this period.

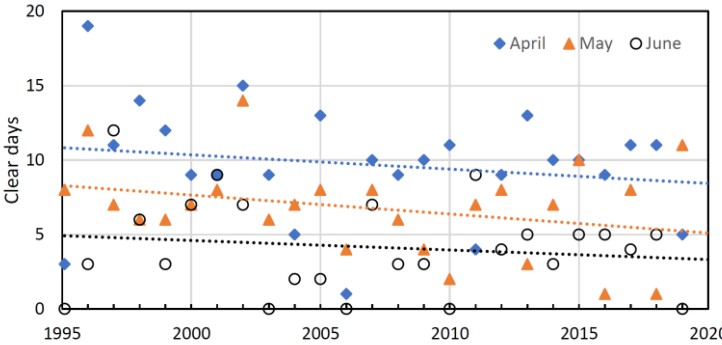

**Figure 14: Number of monthly clear-sky days observed in Ny-Ålesund in April, May, and June 1995 - 2019. Trends are indicated as dotted lines. Data are from the NCCS database.**

**Table 7: Effective cloud transmittance (eCLT) in Ny-Ålesund 1996-2019. "All data" represent monthly noontime average eCLT where all days are included. Clear-sky data represent monthly eCLT noontime average for days with eCLT>100% and classified as clear from the NCCS cloud data. Uncertainty is expressed as 2*STD (2σ).**

| Month | GUV eCLT, all data | | | GUV eCLT, clear-sky data | | |
|---|---|---|---|---|---|---|
| | 1996-2000 avg, % | 2015-2019 avg, % | Trend ± 2σ %/decade | 1996-2000 avg, % | 2015-2019 avg, % | Trend ± 2σ %/decade |
| April | 116.6 | 98.5 | -7.6 ± 4.3 | 125.7 | 115.2 | -4.2 ± 3.6 |
| May | 103.2 | 89.9 | -7.2 ± 3.8 | 122.8 | 114.6 | -3.7 ± 2.4 |
| June | 86.2 | 77.6 | -3.6 ± 5.3 | 114.6 | 109.7 | -2.0 ± 1.3 |

The cloud data from NCCS will partly explain why the overall eCLT in Figure 13 is highest for April and lowest for June. However, the data will not necessarily give the full explanation of the decreasing GUV eCLT trend from 1996 - 2019. To examine whether the decrease in eCLT also is affected by albedo change, clear-sky data (defined as noontime eCLT ≥ 100%) have been selected from the GUV time series and studied separately. These GUV clear-sky data are selected from days where the NCCS cloud data indicate a clear noon, i.e. the sky at 12:00 is classified as category 0, 1, or 2. The results are shown in Figure 15. Note that data from May and June 2005 are missing due to the FARIN calibration campaign (see section 3.2). For June there are also several data gaps in Figure 15 due to the absence of clear-sky days. As seen from Figure 15 there are clear negative eCLT trends for April, May and June also when the effect of clouds has been ruled out.

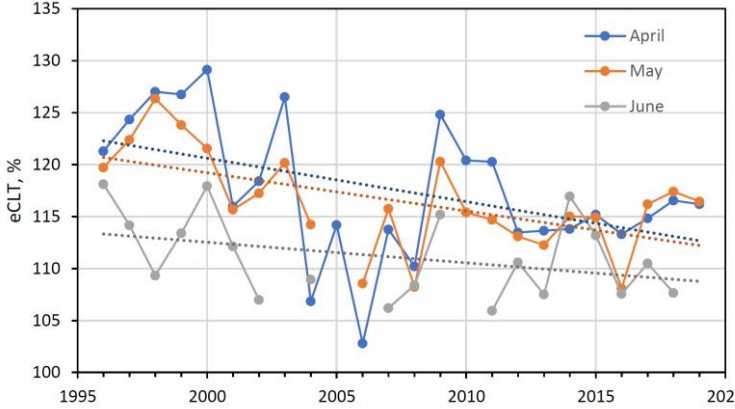

**Figure 15: Monthly mean clear-sky eCLT in Ny-Ålesund for April, May, and June 1996 to 2019. Trends in clear-sky eCLT are indicated as dotted lines.**

Theoretical calculations (Degünther et al., 1998; Degünther and Meerkötter, 2000; Lenoble, 2000) show that surface ultraviolet irradiance measurements may be influenced by albedo variations more than 10-20 km away. Kylling and Mayer (2001) showed that for Tromsø, Norway, a declining snowline in mountainous areas may have about a 25% (50%) effect on cloudless (cloudy) surface irradiance measurements. These findings support the suggestion that the clear-sky eCLT trends in Ny-Ålesund are due to albedo changes. The changes can be attributed to local snow/ice conditions, but also to ice/snow changes several kilometers away from the measuring site.

As seen from Figure 15, there can be large eCLT variations from one year to another. In April 2006 there was a minimum eCLT value of only 103%. As indicated in Figure 14, there was only one clear day in this month (20 April), a day which was classified as category 2 from the NCCS data (a quarter of the sky had clouds). The GUV eCLT minute values indicate that a thin cloud or haze occasionally covered the sun and resulted in relatively low noontime average eCLT this day. April 2009 is an opposite example where the noontime eCLT was very high. This day a large fraction of the NCSS cloud data were classified as category 0, meaning that the sky was cloud free for several days. According to snow data from NCCS, the snow depth in Ny-Ålesund was high in April 2009. In addition, the ice extent in the Barents Sea in spring 2009 was large compared to previous years (Norwegian Polar Institute, 2020). The combined effect of these three factors resulted in a peak eCLT in April 2009.

Clear-sky eCLT mean values and trends from the GUV are summarized in Table 7. The average clear-sky eCLT for April 1996-2000 is 125.7% whereas the April average for 2015-2019 is 115.2%, a decline of ~8% (-4.2 ± 3.6 %/decade). For May there is a similar tendency with decreasing clear-sky eCLT of -3.7 ± 2.4 %/decade. As seen from Table 7, the negative eCLT

trends are significantly reduced for clear-sky data compared to "all data". Whereas the "all data" eCLT is affected by both clouds and albedo, clear-sky eCLT is mainly affected by albedo changes. This indicates that roughly half of the eCLT decline seen in Figure 13 is related to changes in cloud cover, whereas the other half is related to albedo changes. It should be noted that the eCLT decrease seen in Figure 15 do not change significantly if we ignore the NCCS clear-sky selection and only study data with eCLT>100%. This demonstrates that GUV albedo changes can be studied even if independent cloud observations are not available. As mentioned above, aerosols can also influence eCLT in addition to clouds and albedo. However, aerosols in Ny-Ålesund are normally of small importance because of low amounts. Also, no significant aerosol trends have been observed at high latitudes (Eleftheratos et al., 2015).

The eCLT results from Ny-Ålesund imply that there has been a significant change in albedo with reduction of snow/ice in the Svalbard area throughout the last 25 years, especially for the spring months. Related results were found by Bernhard (2011) who showed that the onset of snowfall at Barrow, Alaska, advanced by almost 2 weeks/decade for the period 1991-2011. Also, albedo studies from Möller and Möller (2017) has demonstrated a significant negative albedo trend of the glaciers of Svalbard over the period 1979-2015, and data from the Norwegian Polar Institute shows that the sea-ice extent in April in the Barents Sea has considerably declined the last decades (Norwegian Polar institute, 2020). These findings on Arctic albedo change and ice melt clearly support existing reports and publications on ongoing climate change (Wunderling et al., 2020; IPCC 2018).

## 6. Conclusions

The Norwegian UV network has been in operation for 25 years, and the unique GUV data can be used to derive a broad range of atmospheric and biological exposure parameters, including total ozone column (TOC), UV index, and cloud transmittance. The instruments are relatively simple to operate and maintain and measure continuously throughout the day with 1-minute time resolution.

The 25-year long records of GUV TOC measurements in Norway have been re-evaluated and harmonized. For the three stations located in Oslo, at Andøya and in Ny-Ålesund there are annual TOC increases of 2.3 ± 1.5 %/decade, 1.6 ± 2.2 %/decade, and 2.5 ± 2.2 %/decade, respectively, for the period 1996-2019. However, TOC is strongly influenced by stratospheric circulation and meteorology, and the large interannual variability reduces the statistical significance of the data.

GUV measurements of effective cloud transmittance (eCLT) in Ny-Ålesund, Svalbard, reveal a negative eCLT trend for the spring, indicating that the albedo at this site has decreased over the past 25 years. This is most likely a consequence of an ongoing ice melt caused by increased temperatures in the Svalbard area.

**Data availability**

Harmonized GUV TOC and eCLT data: http://doi.org/10.5281/zenodo.4446609 (Svendby, 2021).

Brewer DS data: https://woudc.org/

SAOZ data: http://www.ndaccdemo.org/

GOME2-A TM3DAM v4.1: http://www.temis.nl/protocols/o3field/overpass_gome2a.php

OMI TM3DAM v4.1: http://www.temis.nl/protocols/o3field/overpass_omi.php

NCCS cloud and snow data: https://klimaservicesenter.no (https://seklima.met.no/observations/)

**Author contribution**

TMS designed the study and performed the analyses. BJ, AD, AK, and GHB performed supporting simulations and analyses. BP and VV provided Brewer#50 data. GHH was responsible for SAOZ data. TMS wrote the paper, and all authors provided input on the paper for revision before submission.

**Competing interests**

The authors declare that they have no conflict of interest.

**Acknowledgement**

We thank the Norwegian Environment Agency for funding total ozone and UV measurements in Oslo/Kjeller, Andøya and Ny-Ålesund. The authors would like to thank Reidar Lyngra at Alomar (Andøya) and staff at the Norwegian Polar Institute for keeping the instruments running. We would also like to thank the two referees for helpful comments and suggestions.

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
