# Peer review of "GUV long-term measurements of total ozone column and effective"

_Atmospheric Chemistry and Physics, 2021_

## Referee Comment (RC2)

Review of the study entitled "GUV long-term measurements of total ozone column and effective cloud transmittance at three Norwegian sites" by Svendby et al.

The work presents long-term measurements of total ozone and effective cloud transmittance from well-maintained ground-based stations in Norway. Efforts to maintain instruments that provide long-term atmospheric measurements deserve recognition and the results of such studies should be published. The study shows the ability of the GUV multi-filter instruments to complement Brewer direct sun total ozone measurements, discusses the dependence of ozone on effective cloud transmittance, and presents the variability and trends in ozone and effective cloud transmittance in Norway. I am in favor of publishing the results of this study without any reservation.

Few minor comments

Line 241 and eq. (4): I see one harmonic in eq. (4) and in figure 1, i.e. the seasonal, and not two harmonics. Please check. If it were two harmonics I would expect the equation to be of the form:

$$f(t) = a + c \cdot \cos(2\pi t) + s \cdot \sin(2\pi t) + c' \cdot \cos(2 \cdot 2\pi t) + s' \cdot \sin(2 \cdot 2\pi t)$$

First harmonic          second harmonic

Equation 5: A quadratic fit would probably fit better as I see from the plots in figure 5, but a linear fit seems also to work. Therefore, all right. It would be good to know what the errors and related statistical significances of the coefficients $a$ and $b$ are, in Table 3.

Line 253 or Lines 295-302: Can you explain what the harmonization procedure offers and how is it used? Is it used as a method to fill gaps in the time series? Is it used as a method to bring closer the GUV data to the Brewer data and vice versa? It is not clear.

Line 490: correct "form GUV" to "from GUV".

Figure 12: I notice that the Average eCLT (April) > Average eCLT (May) > Average eCLT (June). In Figure 13 I notice that the Average overcast days (April) < Average overcast days (May) $\cong$ Average overcast days (June). Shouldn't the Average overcast days in June be larger than May's? Since eCLT is based on noontime values, is the correspondence between the average values in figures 12 and 13 improved if you take cloud observations at 12.00?

Line 491: The cloud data from NCCS represent cloud cover for a whole day. In lines 477-479 you mention that cloud observations are performed three times a day, at 6:00, 12:00, and 18:00. Do cloud observations at 12.00 improve the GUV and NCCS correlation?

Cloud analysis: I understand that clouds have little or no influence on the eCLT trends. Aerosols are probably not important because of low amounts or of no significant trends in high latitudes

(see for instance Eleftheratos, K., Kazadzis, S., Zerefos, C., Tourpali, K., Meleti, C., Balis, D., Zyrichidou, I., Lakkala, K., Feister, U., Koskela, T., Heikkilä, A., and Karhu, J. M.: Ozone and spectroradiometric UV changes in the past 20 years over high latitudes, Atmosphere-Ocean, 53, 117-125, doi: 10.1080/07055900.2014.919897, 2015). As such, I understand that the negative trends in eCLT are attributed to negative trends in surface albedo (less ice coverage) as all other parameters do not explain the observed trends. Is that so?

---

## Author Comment (AC1)

**Reply to referee comments**

ACP-2021-54
Title: GUV long-term measurements of total ozone column and effective cloud transmittance at three Norwegian sites
Author(s): Tove M. Svendby et al.

**Answer to referee #1**

Thank you very much for a detailed review. Your suggestions are highly appreciated and have helped to improve the manuscript. The answers to your comments are in *italic.*

**General comments:**
The manuscript describes both the quality assurance of total ozone column (TOC) and effective cloud transmittance (eCLT) measurements at three Norwegian sites and presents the analysis of the time series. The manuscript is well written and easy to follow, but needs some rearrangement of the text as suggested in the Specific comments. As stated in the Abstract, this work is a good demonstration of how GUV TOC measurements can be used to complete state-of-the art instrumentation. However it is not clear if the methodology used to correct for the seasonal drift is location dependent or not. The approach of using the effective cloud transmittance to assess changes in albedo is interesting, but would benefit from comparison with albedo/snow cover data if such data is available. The data analysis of both TOC and eCLT is more an explanation of observed changes than trend analysis, so I suggest to change the wording from trend analysis to observed changes. I also miss uncertainty estimates of the measurements.

*We have updated the manuscript and included some text about drift correction of the GUVs. Regarding cloud transmittance and albedo, we could not find snow information in the NCCS database prior to 2009. Instead we have looked at the sea ice extent from the Barents Sea (http://www.mosj.no/en/climate/ocean/sea-ice-extent-barents-sea-fram-strait.html) and snow data from selected months that needed a more careful examination (e.g. https://www.yr.no/nb/historikk/graf/5-99910/Norge/Svalbard/Svalbard/Ny-%C3%85lesund?q=2009). For the TOC analyses we have changed the wording from trend to "changes" several places. Finally, more uncertainty estimates have been included in the manuscript.*

**Specific comments:**

**Comment 1:**
Line 49: Generally, a decrease in total ozone column leads to an increase in UVB radiation → That's true assuming no changes in cloudiness (or other UV-affecting parameters). Please update the sentence.

*The sentence was updated*

**Comment 2:**
I suggest to change the title of Chapter 3 to Material and Methods, as you describe also the instrumentation.

*The title was changed*

**Comment 3:**
Please add a brief description of the Brewers and SAOZ in Chapter 3.

*A brief description of Brewer was included. The description of SAOZ was moved from chapter 4 to chapter 3.*

**Comment 4:**
Section 3.2., line 135. Please give a brief description of the methods, including needed equations, for "annual assessments of drift and determination of correction factors".

*The procedure is described in WMO (2008). Also, a separate publication that describes the GUV-network, calibrations and procedures for drift correction is in preparation. We therefore avoided to repeat details but added the following lines:*
*"The irradiance from the two collocated instruments are compared to results from the 2005 calibration campaign, where the drift for all instruments and channels were set to unity. Relative to this 2005 calibration, yearly drift factors $d_i$ for the individual channels and instruments are derived. These drift factors are used to modify the response factor in Eq. (1), $k_i'=k_i/d_i$. If $d_i$ changes from one year to the next, a linear change in $d_i$ is assumed for periods between two inter-comparisons."*

*WMO (2008), Johnsen, B., Kjeldstad, B., Aalerud ,T.N., Nilsen, L.T., Schreder, J., Blumthaler, M., Bernhard, G., Topaloglou, C., Meinander, O., Bagheri, A., Slusser, J.R., Davis, J.:. Intercomparison of global UV index from multiband filter radiometers: Harmonization of global UVI and spectral irradiance. GAW report no. 179 / WMO/TD-No. 1454. Geneve: World Meteorological Organization, 2008.*

**Comment 5:**
Line 149 and eq. (2) : Please explain how the possible drift in V_i or V_j is taking into account, when calculating the total ozone time series. How does the calibration or comparison with the traveling reference is taking into account?

*This is closely related to the previous comment: The annual derived drift factors are used to modify the response factors $k_i$*

**Comment 6:**
Line 168: Could you please justify the choice of 340 nm instead of 380 nm?

*Both channels can be used and give similar results. Thus, it is just a coincidence that the 340 nm channel has been used instead of the 380 nm channel in our study.*

**Comment 7:**
Lines 177-187: I suggest to move this paragraph to Section 3.3. in which you describe the TOC retrieval theory.

*The paragraph was moved*

**Comment 8:**
Lines 195-212: Please move these informations to a separate Section in Material and Methods. Please add information of the uncertainty/error estimates of Brewer and SAOZ TOC.

*The lines were moved to section Material and Methods. Information about Brewer and SAOZ uncertainty was also added.*

**Comment 9:**
Line 217: For the Brewer DS measurements: Which measurements did you use, the daily mean? Or DS around local noon?

*For Brewer DS, the daily means are used. These are the same data as submitted to the WOUDC data base. A comment about this was included in the manuscript.*

**Comment 10:**
Figure 3.: This Figure 3 doesn't convince me. The problem is that in June, the data points stop at SZAs when in April the "spread" or "SZA dependence" only starts. What would this figure look like if you would do it for e.g. Oslo, where also high SZA:s are reached during the summer?

*This is a very relevant question. We have had some internal discussions whether the seasonal correction should be based on "day of year" or SZA. Since we look at noon-time TOC, the two correction methods give almost the same results. As seen from figure 3 (upper panel) there is no clear SZA dependence in the data, but a potential SZA dependence might very well be masked by diurnal ozone variability, clouds, and a general large uncertainty at high SZA. Also, studies of other days/periods with stable weather conditions throughout the day indicate that TOC decreases with increasing SZA. Thus, we have decided to switch to an SZA correction instead of "day of year" correction in Eq. 4, as this might give a better physical explanation of annual variability. For Oslo, the diurnal TOC during summer is different from the pattern seen from other GUVs. In Oslo there is typically a gradual ozone increase from early morning until noon, however, contrary to TOC from the other GUV-541 instruments the values do not necessarily decrease with increasing afternoon SZA. This is explained by the special 305nm filter in the old GUV-511 instrument, which will be influenced by the high solar exposure throughout the day.*

**Comment 11:**
Section 4.1.: I would like to see the effect of the seasonal correction. Please add a Figure similar to Figure 2, but plotted using the applied correction. Consider showing it with data having all corrections applied.

*A new figure with all corrections was added to section 4.2. of the manuscript.*

**Comment 12:**
Section 4.1. Please add statistics of the GUV vs Brewer/SAOZ comparison at all three sites, for non-corrected and for corrected data.

*A table describing the statistics was added in section 4.2.*

**Comment 13:**
Figure 5: For consistency in Fig. 2 and Fig 5., choose weather you prefer to show the results as percentage or as ratio. What is the red continuous line?

*For the CLT correction (Fig 5), ratio is now used instead of percentage. The red continuous line marked the "zero line" for CLT>60%, but this is now removed.*

**Comment 14:**
Line 273: Was the GUV data already corrected for seasonality as shown in Section 4.1.?

*Yes, the GUV data were corrected for the seasonality (SZA) before the CLT correction was established. This is now clarified in the text.*

**Comment 15:**
Line 302: Please quantify the mean bias for all stations.

*A table with bias was included in the manuscript*

**Comment 16:**
Lines 295-302: I suggest to move this info in the Section Data analysis, and show a Figure (see my comment above) including the comparison GUV_all_correction_applied/Brewer.

*Changed as suggested.*

**Comment 17:**
Line 305: GOME-2 data: Please add a brief description in Material and Methods. Is the time series homogeneous?

*A description of GOME2 and OMI data was included in section Material and methods.*

**Comment 18:**
Line 314: How would the albedo affect GOME-2 data? Could Andoya's albedo conditions also affect GUV TOC measurements? I think the GOME-2 TM3DAM data includes some error estimates, have you looked at them?

*Because of its coastal location with steep mountains, the topography, cloud cover, and albedo around Andøya vary on a small scale (< 1 km) that this much smaller than a satellite pixel (> 10 km). Such an environment is challenging for satellite observations. Some lines about this were added to the manuscript.*
*In the updated data set, we have flagged GUV TOC as "uncertain" if GUV ozone retrievals within 1 hour about solar noon vary by more than 20 DU. These measurements have mainly been performed under challenging weather conditions, which have contributed markedly to the high STD between GUV and GOME2 measurements. The STD is reduced significantly when the flagged data are omitted. We have looked at the error estimates from the GOME-2 TM3DAM files but did not find any indications of higher error estimates for Andøya compared to the other stations.*

**Comment 19:**
Table 4: Please include also the mean bias as percentage in addition to DU.

*This was done*

**Comment 20:**
Line 331: "...can be moved to a new location..." Are the corrections / harmonization independent of location? What is the effect of possible differences in the ozone profiles between locations?

*The instrument can be moved to a new location, but if the atmospheric profile is very different, there might be deviations for high SZA. The manuscript was updated to explain this.*

**Comment 21:**
Line 338: Please add a brief description of differences between Brewer DS, ZS and GI measurements in Materials and Methods.

*This is now included*

**Comment 22:**
Line 371: Dynamical transport in the stratosphere?

*The section describing ozone transport has been updated.*

**Comment 23:**
Line 416: Please describe OMI TOC in Material and Methods.

*This was included*

**Comment 24:**
Figure 8: Now that you have included OMI TOC in this plot: why not performing the comparison between GUV and OMI TOC in Section 5.1.? Also when looking at the plot, the question rises, why GOME-2 and OMI differs from each other (and GOME-2 from GUV) after mid-April ?

*A figure with OMI TOC has been included. There are uncertainties in both GUV, GOME2 and OMI data products, and this is briefly described in the text.*

**Comment 25:**
Section 5.2. in general: I am wondering if the word "trend" is the best one to use. For me, the results show more "observed changes", as no trend analysis including the "explanatory terms" mentioned in lines 372-374 is done. Please consider changing the wording.

*The word "observed change" (or similar) will be used instead of "trend" most places. However, the term "trend" is still occasionally used to make the text more readable.*

**Comment 26:**
Lines 442-446. Please check if Oslo was inside the polar vortex on that day. If not, I don't think the explanation is valid, as it's only the start of the chemical ozone loss period. Oslo's latitude is quite low, and the station is not obviously inside the vortex.

From: https://ozonewatch.gsfc.nasa.gov/ozone_maps/movies/OZONE_D2019-11-01%25P1D_G%5e360X240.IOMPS_PNPP_V21_MMERRA2_LNH.mp4
it can be seen that the ozone loss associated to the polar vortex forms later in the season 2019/2020. Could the December 4 case in Oslo be as a result from ozone poor air transport from the mid-latitudes or related to changes in the height of the tropopause / dynamics?

*We have looked at various products (e.g. ERA-5, soundings from Sodankyla) and a recent publication from Dameris et al. (2021) and agree that the "mini ozone hole" probably was a result of advection of lower-altitude airmasses. Thus, we have updated the manuscript with the following text:*

*"In the fall/winter 2019 the Arctic polar vortex formed earlier than usual (Manney et al., 2020, Lawrence et al., 2020). Temperatures were low enough for PSC formation by mid-November 2019, earlier than in any previous year since at least 2004. PSCs were visible over Norway during a large part of the winter 2019/2020. However, in early December, chorine activation and associated chemical ozone loss was still limited. Dameris et al. (2021) indicate that a "mini ozone hole" over Southern Norway on 4 December 2019 was caused by advection of lower-latitude airmasses and increased tropopause height."*

Dameris, M., Loyola, D. G., Nützel, M., Coldewey-Egbers, M., Lerot, C., Romahn, F., and van Roozendael, M.: Record low ozone values over the Arctic in boreal spring 2020, Atmos. Chem. Phys., 21, 617–633, https://doi.org/10.5194/acp-21-617-2021, 2021.

**Comment 27:**
Line 475: Please move the description of cloud data to Materials and Methods.

*This was done*

**Comment 28:**
Line 488-489: It's not clear how you calculated the correlation coefficient. And what was the purpose of the correlation calculation? As eCLT includes the effect of aerosol, albedo and cloudiness, please explain the point of calculating the correlation with cloudiness observations. How about doing it only for seasons with no snow? Please remind the reader of the criteria for the classification "overcast" for GUV data. Also it's not clear in Table 6 that the correlation coefficient is for overcast days.

*I agree that the eCLT and "overcast" correlation is of minor relevance for the study, thus, we have removed this discussion from the text and Table 6. We also found a "new" cloud product in the NCCS data base that describes clear-sky days (at 12:00). In the revised manuscript we have used these clear-sky noon-data instead of overcast days for a full day. Section 5.3 was updated to reflect this change.*

**Comment 29:**
Figure 16: Do you have an explanation for the "jump" or "high" value in 2009. Do you have snow cover data/ ice data which could explain it? Overall, do you have any albedo data / snow cover information which could correlate with this data?

*We have included some sentences about this in the manuscript, especially the eCLT peak in April 2009. This year the ice extent in the Barents Sea was large compared to previous years. Also, the snow depth in Ny-Ålesund was relatively high. Finally, April 2009 was a month with many cloudless days (no small/thin clouds that occasionally covered the sun). The combined effect of these three factors can most likely explain the peak eCLT in April 2009.*

**Comment 30:**
Data availability: Please add cloudiness data from NCCS.

*Cloud data from NCCS has been included*

**Technical corrections:**
Line 60: Brewers spectrophotometers → Brewer spectrophotometers

*This was corrected*

Table 1: For Ny-Ålesund, you have included the serial number. In case you wish to keep it, please add it also to the other stations. If not, then delete it also from Ny-Ålesund.

*The serial numbers were removed from Table 1*

Section 3.3. line 140. Title: Please use the whole words instead of eCLT.

*The title was changed.*

**Answers to referee #2**

Thank you very much for useful and relevant comments to the manuscript. Below are answers (in cursive) to your questions and comments.

**Comment 1:**
Line 241 and eq. (4): I see one harmonic in eq. (4) and in figure 1, i.e. the seasonal, and not two harmonics. Please check. If it were two harmonics I would expect the equation to be of the form:
$(t)=a+c\cdot\cos(2\pi t)+s\cdot\sin(2\pi t)+c'\cdot\cos(2\cdot2\pi t)+s'\cdot\sin(2\cdot2\pi t)$
First harmonic second harmonic
Equation 5: A quadratic fit would probably fit better as I see from the plots in figure 5, but a linear fit seems also to work. Therefore, all right. It would be good to know what the errors and related statistical significances of the coefficients $\alpha$ and $b$ are, in Table 3.

*Thank you for pointing out the issue about harmonics, however, this equation and corresponding explanation have been removed from the revised manuscript. We have previously used a "day of year" correction of our GUV TOC values based on Eq. (4), but after discussions motivated by Reviewer #1, we decided to use a SZA correction instead, as this gives a better physical explanation of the annual variability. When we look at noon-time TOC, the two correction methods give almost the same results. In the revised manuscript, Figure 2 (left panel) illustrates the SZA dependence. The new coefficients are listed in Table 3. We have also included error estimates of the coefficient in Eq. (4) and Eq. (5).*

**Comment 2:**
Line 253 or Lines 295-302: Can you explain what the harmonization procedure offers and how is it used? Is it used as a method to fill gaps in the time series? Is it used as a method to bring closer the GUV data to the Brewer data and vice versa? It is not clear.

*The harmonization procedure (regardless whether it is based on SZA or time) is used to minimize small systematic errors in GUV TOC data and assumes that Brewer data are without error. A sentence about this has been included in the manuscript.*

**Comment 3:**
Line 490: correct "form GUV" to "from GUV".

*This has been corrected*

**Comment 4:**
Figure 12: I notice that the Average eCLT (April) > Average eCLT (May) > Average eCLT (June). In Figure 13 I notice that the Average overcast days (April) < Average overcast days (May) ⏵ Average overcast days (June). Shouldn't the Average overcast days in June be larger than May's? Since eCLT is based on noontime values, is the correspondence between the average values in figures 12 and 13 improved if you take cloud observations at 12.00?

*We found a "new" cloud product in the NCCS database that allows extracting cloud information at 12:00 and not only the average for a full day. Instead of looking at overcast days, we are now using clear-sky data (at 12:00). This means that Figure 13 (Figure 14 in the revised manuscript) is replaced with a new figure showing the number of clear-sky days every month. These data show that it was generally more cloudless days in the late 1990s than today. Also, there are generally more cloudless days in April compared to May, and more in May than June. This will partly explain the overall trend*

*patterns in Figure 12 (now Figure 13). The text in Section 5.3 has been updated to explain this better. The new figures should answer your questions about clouds and eCLT. Normally we would expect Average eCLT (May) > Average eCLT (June) because albedo is larger in May.*

**Comment 5:**
Line 491: The cloud data from NCCS represent cloud cover for a whole day. In lines 477-479 you mention that cloud observations are performed three times a day, at 6:00, 12:00, and 18:00. Do cloud observations at 12.00 improve the GUV and NCCS correlation?

*In the original NCCS cloud dataset it was only one value per day (overcast or not overcast), where cloud information from 6:00, 12:00 and 18:00 were already combined and integrated. As mentioned above, we have introduced a new dataset where cloud information at 12:00 could be retrieved separately. Thus, we decided to use these data instead of the full-day data, as it more relevant for our noon-time study. We also removed the correlation from Table 7 (right column), as Reviewer #1 commented that this was confusing and of less relevance for the publication.*

**Comment 6:**
Cloud analysis: I understand that clouds have little or no influence on the eCLT trends. Aerosols are probably not important because of low amounts or of no significant trends in high latitudes
(see for instance Eleftheratos, K., Kazadzis, S., Zerefos, C., Tourpali, K., Meleti, C., Balis, D., Zyrichidou, I., Lakkala, K., Feister, U., Koskela, T., Heikkilä, A., and Karhu, J. M.: Ozone and spectroradiometric UV changes in the past 20 years over high latitudes, Atmosphere-Ocean, 53, 117-125, doi: 10.1080/07055900.2014.919897, 2015). As such, I understand that the negative trends in eCLT are attributed to negative trends in surface albedo (less ice coverage) as all other parameters do not explain the observed trends. Is that so?

*When we studied clear-sky data at 12:00 it indicated that clouds most likely can explain parts of the large eCLT trend in Ny-Ålesund during spring. However, aerosols have probably insignificant relevance. We have included some text about aerosols in the manuscript, with reference to Eleftheratos et al., 2015.*